# Statin therapy inhibits fatty acid synthase via dynamic protein modifications

Alec G. Trub[1,2], Gregory R. Wagner[1,3], Kristin A. Anderson[1,2], Scott B. Crown [1], Guo-Fang Zhang [1,3], J. Will Thompson[2,4], Olga R. Ilkayeva [1,3], Robert D. Stevens[1], Paul A. Grimsrud [1,3], Rhushikesh A. Kulkarni[5], Donald S. Backos [6], Jordan L. Meier [5] & Matthew D. Hirschey [1,2,3✉]

Statins are a class of drug widely prescribed for the prevention of cardiovascular disease, with pleiotropic cellular effects. Statins inhibit HMG-CoA reductase (HMGCR), which converts the metabolite HMG-CoA into mevalonate. Recent discoveries have shown HMG-CoA is a reactive metabolite that can non-enzymatically modify proteins and impact their activity. Therefore, we predicted that inhibition of HMGCR by statins might increase HMG-CoA levels and protein modifications. Upon statin treatment, we observe a strong increase in HMG-CoA levels and modification of only a single protein. Mass spectrometry identifies this protein as fatty acid synthase (FAS), which is modified on active site residues and, importantly, on non-lysine side-chains. The dynamic modifications occur only on a sub-pool of FAS that is located near HMGCR and alters cellular signaling around the ER and Golgi. These results uncover communication between cholesterol and lipid biosynthesis by the substrate of one pathway inhibiting another in a rapid and reversible manner.

---

[1] Sarah W. Stedman Nutrition and Metabolism Center, Duke Molecular Physiology Institute, Durham, NC, USA. [2] Department of Pharmacology & Cancer Biology, Durham, NC, USA. [3] Division of Endocrinology, Metabolism, and Nutrition, Department of Medicine, Durham, NC, USA. [4] Duke Proteomics and Metabolomics Shared Resource, Duke University Medical Center, Durham, NC 27710, USA. [5] Chemical Biology Laboratory, Center for Cancer Research, National Cancer Institute, National Institutes of Health, Frederick, MD 21702, USA. [6] Computational Chemistry and Biology Core Facility, Skaggs School of Pharmacy and Pharmaceutical Sciences, University of Colorado, Anschutz Medical Campus, Aurora, CO 80045, USA. ✉email: matthew.hirschey@duke.edu

Statins are one of the most widely prescribed drug classes in the world, due to their effective prevention of cardiovascular events. Recent changes in guidelines may lead to as many as 56 million adults in the U.S. being eligible for statin therapy, making research into statin effects particularly relevant[1]. Statins inhibit HMG-CoA reductase (HMGCR) in the cholesterol bio-synthesis pathway, and upregulate genes involved in cholesterol uptake. The increased production of the LDL receptor and con-sequent reduction in blood cholesterol is considered the main mechanism driving the beneficial effects of statins[2]. Statins also exhibit pleiotropic beneficial effects on platelet function, vascular endothelium, and smooth muscle[3]. However, statins have known side-effects including muscle soreness or "statin myopathy", rhabdomyolysis (a severe form of myopathy), and increased risk of new-onset type 2 diabetes[4]. How statins exhibit cellular effects beyond HMGCR is under intense investigation.

The acylation of lysine residues on proteins is a well described post-translational modification that can impact the function of modified proteins[5–7]. These modifications occur primarily via enzymatic transfer but recently non-enzymatic acylation has been described. Acyl-CoAs that contain a four- or five-carbon back-bone undergo self-hydrolysis and form a reactive intermediate, which can readily modify proteins[5]. Due to this chemical prop-erty, the increased abundance of specific CoA species can lead to increased protein modification. In a model of HMG acid uremia, knocking out hydroxymethylglutaryl-CoA lyase (*Hmgcl*) in mice led to increased modification of proteins with HMG (i.e., "HMGylation") and impacted the function of enzymes in the citric acid cycle[5].

As statins inhibit HMGCR which consumes HMG-CoA, we predicted that statin inhibition of HMGCR would lead to increased levels of HMG-CoA and would increase protein HMGylation. We show that statin treatment induces HMGylation on a subpool of fatty acid synthase (FAS) in proximity of HMGCR, affecting cellular signaling and suggesting a statin-induced crosstalk between lipid and cholesterol biosynthesis.

## Results

**Statins induce protein HMGylation on fatty acid synthase.** We first tested if HMG-CoA levels increased during statin-treatment. HepG2 cells were treated with simvastatin for 24 h and acyl-CoA species were analyzed using targeted mass spectrometry. Out of 10 acyl-CoAs measured, only HMG-CoA levels increased ($\sim$3.5-fold, $p < 0.001$, Fig. 1a). The change of this singular CoA species showed inhibiting HMGCR with statins leads to a specific and potent increase of its substrate.

As we previously showed HMG-CoA can spontaneously modify several proteins[5], we next examined if increases in HMG-CoA resulted in protein HMGylation. We incubated HepG2 cells with increasing concentrations of simvastatin across several time points. Using our previously described antibody recognizing HMG-lysine[5], western blotting revealed a singular band of HMGylation occurring at all time points and concentrations (Fig. 1b). Increases in HMGCR confirmed the effects of simvastatin on HepG2 cells and suggested HMGylation is an effect of HMGCR inhibition (Fig. 1b). Remarkably, the single band of HMGylation was present at the earliest time point showing the modification occurred rapidly following the inhibition of HMGCR.

The presence of a single HMGylation band was surprising because we previously knocked-out the HMG-CoA consuming enzyme HMG-CoA lyase, which led to a broad increase in HMGylation[5]. Similarly, lysine acylation western blots and proteomic experiments typically reveal tens to hundreds of modified proteins. The single band of HMGylation allowed for a targeted follow up to identify the HMGylated protein. Performing an immunoprecipitation (IP) on statin-treated cell lysate using anti-HMG-lysine antibody, followed by mass spectrometry revealed a 6.22 fold difference in fatty acid synthase (FAS) between control and HMG IP (Fig. 1b and Supplementary Fig. 1). FAS is a 273 kDa protein that synthesizes long chain fatty acids, suggesting an unexpected link between fatty acid and cholesterol biosynthesis.

Next, we extended these cellular observations to an animal model. C57BL/6NJ mice were fed a chow diet containing atorvastatin, a commonly selected statin for in vivo studies, for 2 to 4 weeks. Again, we first tested hepatic acyl-CoA levels and found increases in HMG-CoA, but no other acyl-CoAs during statin treatment ($p < 0.001$, Fig. 1c), consistent with statin-treated cells (Fig. 1a).

We performed western blotting to see if the same singular band of HMGylation was observed within the livers of statin-treated mice. These blots revealed similar results to those from statin-treated cells, with a single band at the same mass as FAS (Fig. 1d). We observed variability in the level of HMGylation for each mouse, not just in the representative experiment shown but also in the repeat experiments. This variability was an early indicator for the reversibility of the modification, which will be discussed in detail shortly. Together, these data reveal statins induce HMGylation of FAS in HepG2 cells and mice following an increase in HMG-CoA.

**HMGylation of FAS is independent of HMG-anhydride for-mation and relies on CoA.** Our prior work on HMGylation showed the non-enzymatic modification of proteins by the formation of a reactive HMG anhydride from the CoA species. While this mechanism was a rationale for our work with statins, the specific modification of a singular protein indicated that another mechanism might be playing a role. The non-targeted mechanism of modifications by an anhydride would suggest that multiple proteins in the subcellular location would be HMGylated.

We first confirmed that we could recapitulate the modification in vitro by incubating HepG2 cell lysate with HMG-CoA. This rapidly resulted in the specific modification of FAS, just as observed in statin-treated cells and mice (Fig. 2a). Next, we denatured the cell lysate with the addition of 1 % SDS prior to incubating it with HMG-CoA. Comparison of non-denatured, and SDS treated lysate revealed that FAS only became HMGylated when the lysate was non-denatured (Fig. 2b), which indicated that the structural confirmation of FAS was necessary for the HMGylation event to take place. This is not what we would expect to find from a mechanism that relies on the formation of an anhydride, because the reactivity that drives the reaction is found on the metabolite, not derived from the targeted enzyme. The requirement of the proper confirmation of FAS indicates that its structure plays a role in the specificity of the HMG modification.

We further tested this by synthesizing an HMG-CoA-derivative called 5-(2-acetamidoethylthio)−3-hydroxy-3-methyl-glutaric acid (HMG-NAC, Supplementary Figs. 2–4). A similar acyl-NAC compound was previously used to investigate the occurrence of malonyl modifications, and whether this modifica-tions relied on the CoA metabolite to occur[8]. Similar to HMG-CoA, HMG-NAC has a thioester bond that facilitates HMG anhydride formation, however, it does not have a structural similarity to CoA, so any modifications that rely on the CoA moiety will not occur with HMG-NAC treatment. When we incubated HepG2 cell lysate with HMG-NAC, we found that only HMG-CoA was able to form the HMG-modification on FAS (Fig. 2c). These results demonstrate that the CoA moiety of

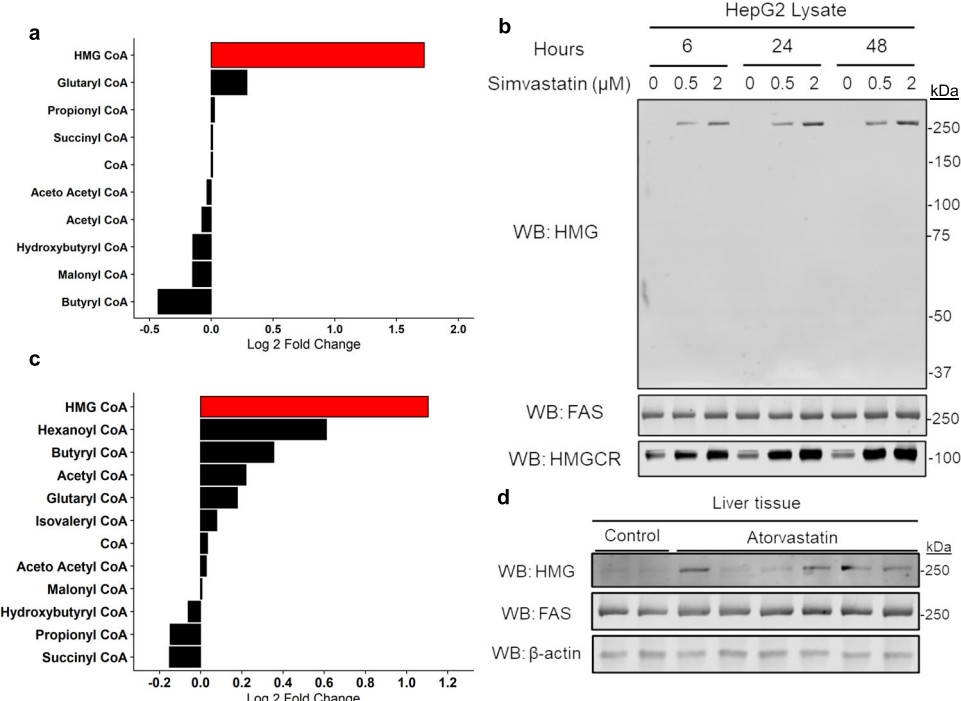

**Fig. 1 Statins induce protein HMGylation on fatty acid synthase.** Statins raise HMG-CoA levels in cells and mice resulting in HMGylated FAS. **a** Treating HepG2 cells for 24 h with simvastatin results in increased HMG-CoA levels with no change to other measured CoAs ($p < 0.001$, $n = 6$ from two independent experiments each with three biological replicates, ANOVA followed by Tukey post-hoc test). **b** Western blotting reveals a singular band of simvastatin-induced HMGylation in HepG2 cells (representative blot of three independent experiments). **c** C57/BL6NJ mice treated with an atorvastatin diet led to an increase in HMG-CoA levels ($p < 0.001$, $n = 4$ biological replicates, ANOVA followed by Tukey post-hoc test). **d** Western blotting of mouse liver shows a singular band of HMGylation following 2 weeks of atorvastatin treatment (representative blot of three independent experiments). Source data are provided as a Source Data file.

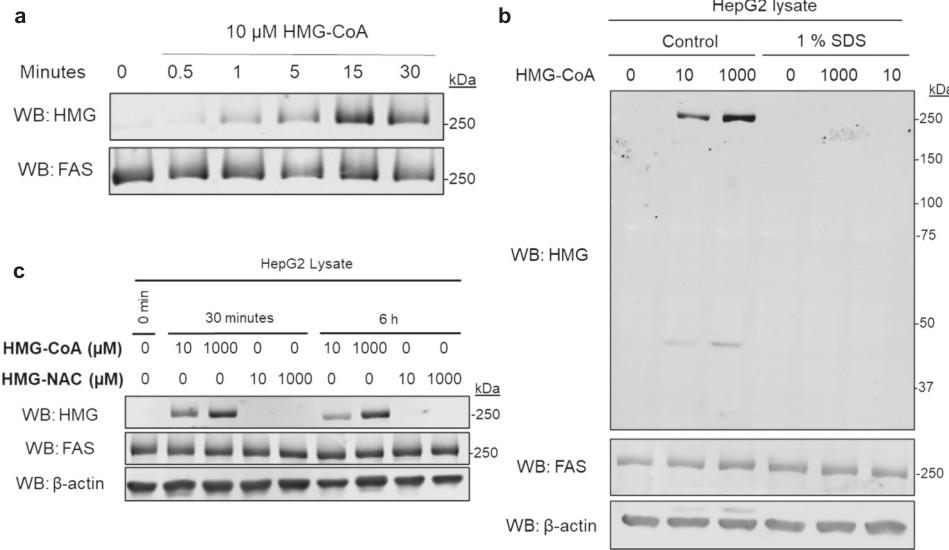

**Fig. 2 HMGylation of FAS is independent of HMG-anhydride formation and relies on CoA.** The HMGylation of FAS is dependent on the CoA handle of HMG-CoA and does not rely on HMG anhydride formation as predicted. **a** HMG-CoA was incubated at room temperature with HepG2 lysate and incubated for up to 30 min revealing the ability for FAS to be HMGylated in vitro. **b** HepG2 lysate with or without SDS to denature FAS followed by incubation with HMG-CoA for 30 min. HMGylation only occurred in the non-denatured lysate, indicating that the structural confirmation of FAS is an important component to its HMGylation. **c** HepG2 lysate was treated with either HMG-CoA or HMG-NAC. Although both HMG-CoA and HMG-NAC can facilitate the formation of the protein modifying HMG anhydride, only HMG-CoA was able to HMGylate FAS showing that the CoA moiety is a requirement for the HMGylation of FAS. Western blots are representative of three independent experiments. Source data are provided as a Source Data file.

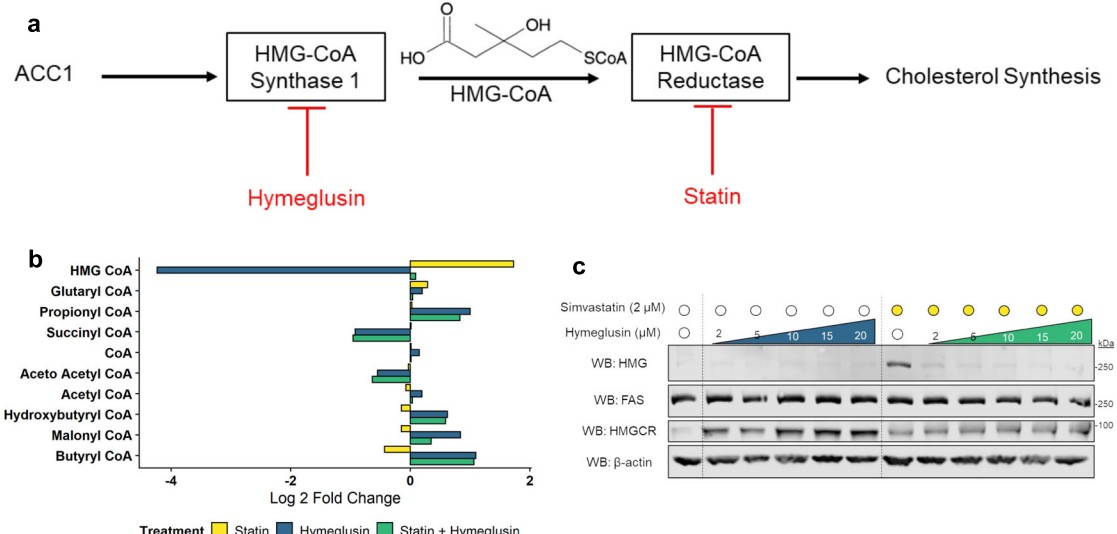

**Fig. 3 HMGylation is dependent on the cholesterol synthesis pathway.** Inhibition of the cholesterol biosynthetic pathway upstream of HMGCR prevents HMGylation of FAS. **a** Hymeglusin inhibits the cholesterol biosynthetic pathway at HMGCS1. This enzyme is responsible for synthesizing HMG-CoA and is directly upstream of HMGCR. **b** Co-treatment of HepG2 cells with hymeglusin and simvastatin results in no change of HMG-CoA from baseline ($n = 3$ biological replicates, $p = 0.99$). **c** Western blotting reveals no HMGylation change from baseline during hymeglusin treatment (representative blot of three independent experiments). Co-treatment of cells with hymeglusin and simvastatin prevent the HMGylation of FAS typically seen during statin treatment. Source data are provided as a Source Data file.

HMG-CoA is a requirement for FAS to be modified and is part of the mechanism that gives specificity to the HMGylation of FAS during statin treatment.

**HMGylation is dependent on the cholesterol biosynthesis pathway**. To determine if the modification observed on FAS originated from the cholesterol biosynthetic pathway, we used a pharmacological tool called hymeglusin[9] to inhibit HMG-CoA Synthase 1 (HMGCS1), the enzyme that generates HMG-CoA for HMGCR (Fig. 3a). The inhibition of HMGCS1 with hymeglusin during statin-treatment prevented the increase in HMG-CoA levels seen previously (Fig. 3b). These results indicate that the majority of HMG-CoA detected by mass spectrometry is being used by HMGCR; further, increased levels of HMG-CoA observed during statin-treatment are directly linked to the cholesterol synthesis pathway. Finally, we looked to see how hymeglusin treatment impacted HMGylation of FAS. Just as the increase in HMG-CoA was prevented by hymeglusin-statin co-treatment, the HMGylation of FAS was also prevented by co-treatment (Fig. 3c). These results demonstrate that flux through the cholesterol biosynthetic pathway directly modulates the levels of HMG-CoA and HMGylation of FAS.

**HMGylation on FAS is a dynamic and reversible modification**. The unusual specificity of a single HMG-modified protein led us to characterize this new modification further. First, we created an in vitro system using purified endogenous *Sus scrofa domesticus* FAS. Next, we incubated purified protein with HMG-CoA, which resulted in its HMGylation. Using this system, we were able to explore the chemical behavior of the HMG modification on FAS. We started by investigating how the HMGylation of FAS was influenced by other metabolites. Relatively low concentrations of HMG-CoA potently modified purified FAS. However, we found the HMG signal could be lowered by adding other metabolites, including malonyl-CoA, a canonical substrate of FAS. Specifically, a fixed amount of HMG-CoA (10 µM) was added to purified FAS while various amounts of malonyl-CoA were added. We found

that as the concentration of malonyl-CoA increased, HMGylation of FAS decreased until it was no longer detectable above baseline levels (Fig. 4a). The gradual decline in HMGylation demonstrated that malonyl-CoA competes with the ability for HMG-CoA to modify FAS.

Furthermore, pre-incubating FAS with HMG-CoA to induce HMGylation does not change the ability of malonyl-CoA to reduce the modification signal from FAS (Fig. 4b). Purified FAS was mixed with 10 µM HMG-CoA for 30 min before an equal amount of malonyl-CoA was added to the mixture and aliquots taken at the indicated times points (Fig. 4b). These results show that in as little as 5 min, an equal molar amount of malonyl-CoA decreased the amount of HMGylation present on FAS.

Owing to the rapid induction and removal of HMGylation in vitro, we returned to a cell culture system to measure how quickly statins impact HMGylation. When HepG2 cells were treated with simvastatin we found HMGylation was induced within 15 min of treatment (Fig. 4c). Importantly, these results indicate that HMGylation is rapidly induced and does not require prolonged statin treatment to occur.

Next, we investigated the reduction of HMGylation following removal of simvastatin. After HepG2 cells were treated for 24 h with simvastatin, media was removed and cells were washed with PBS, before treatment free media was added. Thirty minutes following the removal of statins, HMGylation of FAS was diminished (Fig. 4d). The level of HMGylation appeared be approximately equal at all time points measured following statin removal, although the level does not appear to return to the level of non-treated cells (Fig. 4d). This indicates that the modification is rapidly removed following the withdrawal of simvastatin and is in flux during statin treatment. The ability for this modification to be out-competed or removed by malonyl-CoA, as well as the disappearance of the modification following statin withdrawal, indicates it is highly dynamic, unlike other acyl-based post-translational modifications.

Serendipitously, we noticed that the modification only appeared when western blotting was performed without heating or reducing agents, a key factor when blotting for HMGCR. When we directly

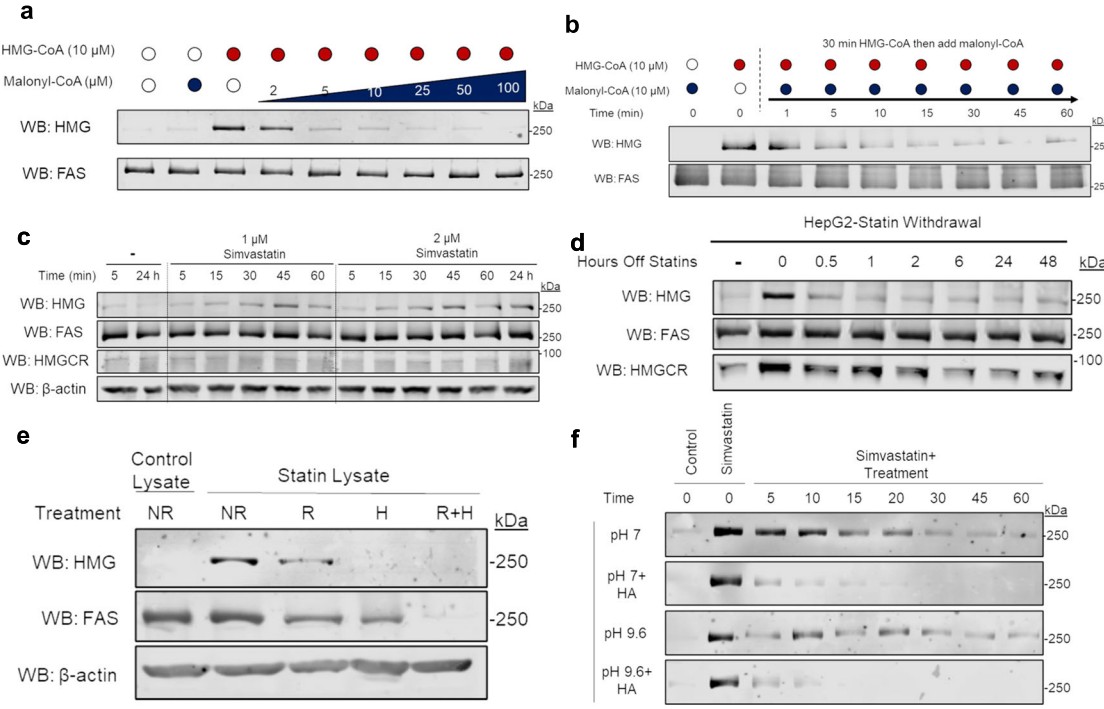

**Fig. 4 HMGylation on FAS is a dynamic and reversible modification.** The modification of FAS is a labile modification that can be rapidly induced or removed and the chemical properties of the modification indicate it is not a typical lysine-based acyl-PTM. **a** Western blotting shows the incubation of purified FAS with malonyl-CoA prevents the HMGylation of FAS typically observed when incubated with HMG-CoA. This occurs to a greater extent the higher the ratio of malonyl-CoA to HMG-CoA. **b** Purified FAS that has been HMGylated by incubation with HMG-CoA, loses the modification when an equal molar amount of malonyl-CoA is added. **c** Western blotting shows HMGylation of FAS in HepG2 cells is induced within an hour following statin-treatment. **d** The HMGylation signal within HepG2 cells diminishes within 30 min following the removal of statins from the media. **e** The addition of DTT (final 100 mM) to the western blot samples results in diminished HMGylation signal via western blot, while the boiling of samples completely removes HMGylation. **f** Hydroxylamine at pH 7 causes the HMG modification to diminish in lysates from statin-treated cells, indicating the HMG modification is linked via a thioester bond. The stronger pH 9.6 hydroxylamine targets both ester and thioester bonds and the total loss of signal supports that the HMG modification is linked via a thioester and ester bond. Western blots are representative of three independent experiments. Source data are provided as a Source Data file.

tested these factors, we found that heating completely abolished the HMGylation signal, while the reducing agent DTT lowered the signal (Fig. 4e). This behavior is unlike other acyl modifications previously described, which tend to be stable unless enzymatically removed. Indeed, most acyl-PTMs are covalently bound to lysine residues and the amide bond that is formed is highly stable. Thus, we considered that HMG might be bound to FAS through amino acids other than lysine.

We used a chemical treatment to characterize the bonds formed between HMG and FAS. We treated HMG-modified FAS with hydroxylamine (HA) at pH 7 or pH 9.6. At pH 7, HA cleaves thioester bonds (e.g., HMG-cysteine or other thiol), while at pH 9.6 cleaves both thioester and ester bonds (e.g., HMG-cysteine + HMG-serine/threonine/tyrosine)[10]. When mixed with lysate from statin-treated cells, HA at pH 7 resulted in the slow loss of signal, indicating a major portion of the signal is from a thioester bond. Additionally, treatment using pH 9.6 HA resulted in rapid loss of signal (Fig. 4f and Supplementary Fig. 5), revealing that the modification is comprised of both thioester and possibly ester bonds.

**HMGylation occurs on FAS active site residues**. We sought to identify the exact site(s) of HMGylation via liquid chromatography–mass spectrometry (LC–MS/MS). In particular we were interested in known active site residues that would form these bonds and sought to optimize a protocol that would (1) maintain the chemically sensitive modification and (2) provide coverage of these sites. We first used purified FAS mixed with or

without HMG-CoA to provide clear positive and negative samples. We followed this experiment using FAS IP'd from control and statin-treated HepG2 cells. In both experiments we identified the prosthetic group phosphopantetheine as being HMGylated via a thioester bond (Fig. 5a, c, d). We were able to confirm the modified phosphopantetheine by using Parallel Reaction Monitoring (PRM) and measuring the predicted ejection ion[11]. These results provide evidence that HMGylation occurs on a prosthetic group of FAS, which is directly involved in the catalysis of fatty acids.

Following on these studies, we used non-targeted LC-MS/MS to identify HMGylated serine 581, which is the portion of the active site on FAS responsible for accepting acyl groups (Fig. 5b, c, d). The selected spectrum conclusively shows the HMG modification on serine 581 and we additionally identified other MS/MS spectra, with different sequence overlap from the pepsin digestion. These corroborate the HMG localization albeit with less localization specificity than the spectrum shown. While this residue was not identified within the statin-treated cells, its absence is likely due to the reduced efficiency of the protocol, the labile nature of the modification, and the longer prep time associated with collecting protein from cells. The final HMGylated residue identified was lysine 673 (Fig. 5d). Little is known about this residue's function although it was previously identified as HMGylated in mice[5] and is in close proximity to serine 581. This lysine residue could explain the chemically resistant HMG signal observed by western blotting. Together, these data show modification of multiple active sites and opens the possibility to further acyl-modifications on non-lysine residues.

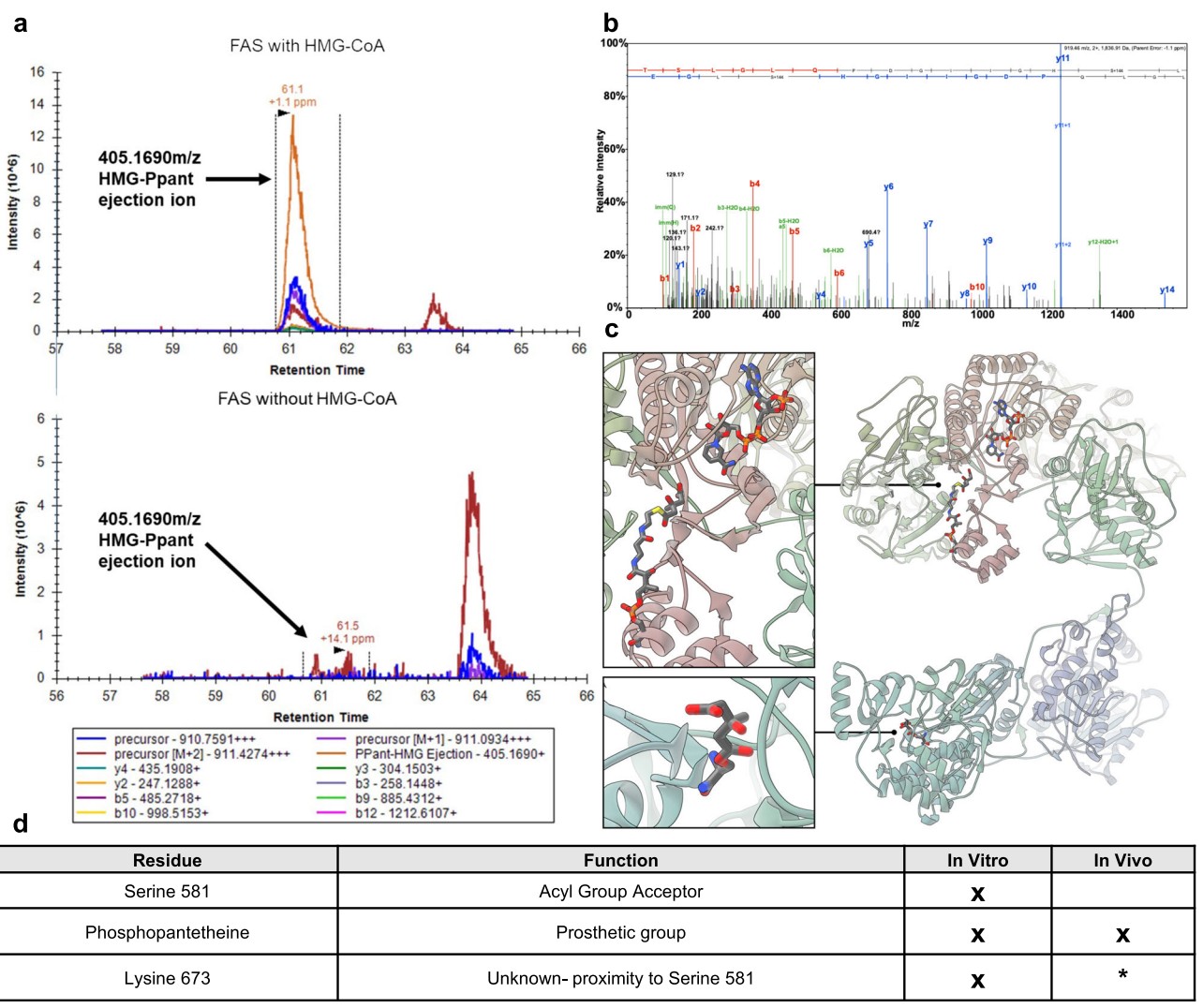

**Fig. 5 HMGylation occurs on FAS active site residues. a** An extracted ion chromatogram revealing the phosphopantetheine ejection ions from in vitro samples with FAS incubated with HMG-CoA (top) and not incubated with HMG-CoA (bottom). The indicated peak corresponds with the predicted ejection ion of HMGylated-Phosphopanetheine. **b** Mass spectrum from Pepsin digest of HMGylated revealing serine 581 is a site of HMGylation. **c** Modeled image showing the HMGylation sites on FAS active site residues. Phosphopantetheine (top) and serine 581 (bottom). **d** Table showing which HMGylated residues were detected by each mass spectrometry experiment. *HMGylated lysine was previously detected in mouse liver although not identified in our cell experiments[5]. Source data are provided as a Source Data file.

**HMGylation inhibits a subpool of FAS and impacts cellular signaling**. The discovery of the modification existing on multiple active sites, prompted the next step of identifying an impact on the activity of FAS. Purified FAS was pre-incubated with HMG-CoA before the addition of all required substrates for activity. The oxidation of NADPH was monitored via UV–Vis spectroscopy and the slope of the 340 nm absorbance was used for comparing relative activity between HMG-CoA containing samples and control samples. The addition of HMG-CoA led to a reduction in activity of FAS, with activity plateauing at ~10% of control (Fig. 6a). The reduction in activity appears related to the amount of HMG-CoA; interestingly, at equimolar concentrations of HMG-CoA and malonyl-CoA the activity is approximately halved (Fig. 6a).

The major role for FAS has been described as producing palmitate to use in the formation of triglycerides for energy storage. To test whether this fatty acid synthesis was impacted in vivo, we first examined palmitate production in statin-treated HepG2 cells. Cells were cultured with or without simvastatin in media containing 10 % deuterated water. Analysis of cell lysates

via GC-MS allowed us to measure de novo *lipogenesis* by detecting the incorporation of deuterium in newly synthesized palmitate. Comparison of newly synthesized palmitate revealed no differences in the fractional synthesis rate of palmitate between control and statin-treated cells (Fig. 6b and Supplementary Fig. 6), consistent with no known effects of statins on lipid biosynthesis.

We next tested lipid biosynthesis in a mouse model. Mice were treated with atorvastatin in chow for 2 weeks, and deuterated water was introduced 48 h prior to liver collection for lipid measurements. We found no difference in either the % labeled palmitate or total palmitate between the statin and control samples (Fig. 6c and Supplementary Fig. 7a). To be sure that other lipids were not affected, both stearate and myristate were measured and no differences found during statin treatment (Supplementary Fig. 7b). Thus, consistent with cell culture results, these data showed no change in de novo *lipogenesis* upon statin treatment, despite potent reductions in FAS activity.

One explanation for the conflicting in vitro and in vivo data on the effects of HMGylation on FAS could be that the HMGylation

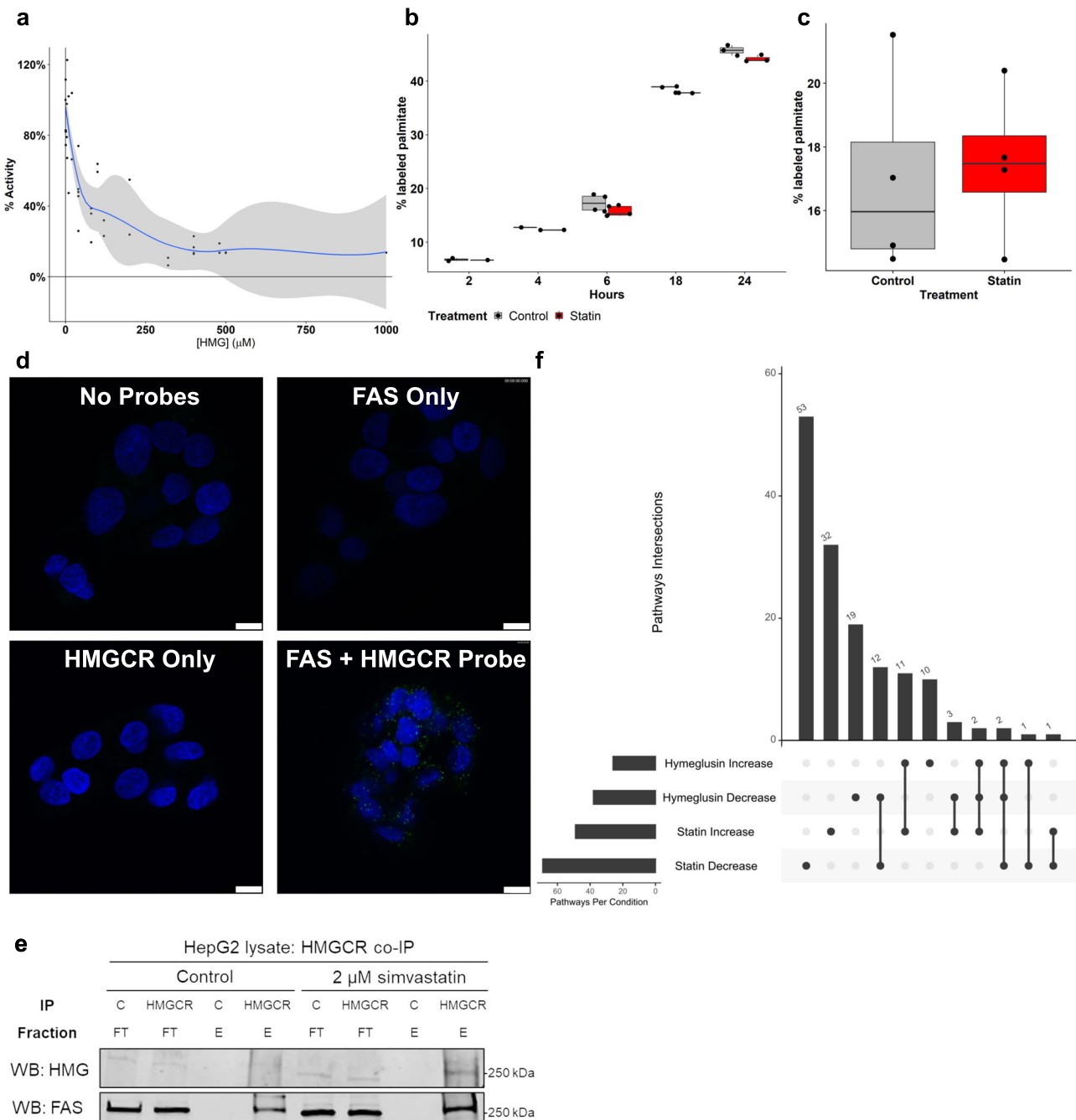

**Fig. 6 HMGylation inhibits a subpool of FAS and impacts cellular signaling.** The HMGylation of FAS results in decreased activity in vitro but without changes in palmitate production in cells or mice. A subpool of FAS interacts with HMGCR possibly impacting cellular signaling. **a** In vitro activity of FAS as measured by the oxidation of NADPH. Activity is a percentage of FAS activity when no HMG-CoA is present. Increasing concentrations of HMG-CoA in the assay result in decreased FAS activity ($p < 0.001$, $n = 6$, three independent experiments with two technical replicates each. Locally weighted smoothing (loess) was used to plot a regression line showing the relationship between % activity and the concentration of HMG-CoA; gray shaded error bands are standard error (se) confidence intervals. **b** HepG2 cells labeled with deuterated water show no difference in the synthesis rates of palmitate between control cells and cells treated with simvastatin ($p = 0.25$, $n = 3$ biological replicates, ANOVA; raw data is shown as points while a best-fit boxplot displays the median with two hinges corresponding to the first and third quartiles (the 25th and 75th percentiles), and two whiskers; the upper whisker is 1.5 * IQR from the hinge (where IQR is the inter-quartile range, or distance between the first and third quartiles), while the lower whisker extends from the hinge to the smallest value at most 1.5 * IQR of the hinge. Data beyond the end of the whiskers are "outlying" points plotted individually). **c** Deuterium labeling in the livers of mice does not reveal a difference in the rate of palmitate synthesis during statin-treatment ($p = 0.8$, $n = 4$ biological replicates, ANOVA, boxplot as in **b**). **d** Proximity ligation assay was performed on HepG2 cells and sites of HMGCR-FAS interaction were confirmed by the presence of green puncta. Negative controls withhold one or both probes and reveal no green puncta (representative of three independent experiments, each with two biological replicates; scale bar is 10 μm). **e** Co-IP of FAS reveals that FAS precipitates with HMGCR regardless of statin treatment. Additionally, the FAS that is associated with HMGCR is HMGylated during statin treatment. *C = Control IP with protein A only; HMGCR = HMGCR IP; FT = flow through fraction from IP; E = eluate from IP. **f** Upsetter plot showing the treatment intersections of enriched pathways from the label-free quantitative proteomics experiment. Source data are provided as a Source Data file.

signal found in statin-treated cells comes from a localized source of FAS that does not contribute to the bulk production of palmitate. Recent studies have shown that several pools of FAS exist within different compartments of the cell, are differentially regulated, and affect the activity of proteins they are associated with beyond bulk lipid biosynthesis[12,13].

Owing to the unexpectedly specific HMGylation of FAS, we hypothesized that a subpool of modified-FAS would be near the site of inhibition at HMGCR embedded in the endoplasmic reticulum (ER). Therefore, we used a proximity ligation assay to determine whether FAS was directly interacting with HMGCR. While the omission of either probe resulted in no observed interactions, multiple instances of FAS and HMGCR interactions appeared when all components were present (Fig. 6d). This result supports the notion that a fraction of FAS is located near HMGCR and is positioned to be HMGylated upon statin treatment.

We further confirmed this interaction by performing a co-immunoprecipitation of HMGCR. HepG2 cells were treated with control media or 2 μM simvastatin before HMGCR was immunoprecipitated and western blotting performed to determine the presence of FAS. This revealed that FAS co-precipitates with HMGCR, confirming this previously unreported interaction (Fig. 6e). Additional blotting revealed that not only does FAS co-IP with HMGCR, but this portion of FAS is HMGylated, supporting our hypothesis that this interaction drives the modification of a specific pool of protein (Fig. 6e and Supplementary Fig. 8). Together the proximity ligation, and the co-IP experiments show that a subpool of FAS interacts with HMGCR and this interaction enables the modification of a specific subpool of FAS.

Prior work has shown subpools of FAS to be associated with a myriad of cellular processes. In order to identify any cellular changes that occur due to the HMGylation of this subpool of FAS, we performed label-free quantitative proteomics to compare protein changes between treatment conditions. HepG2 cells were treated with either vehicle, simvastatin, or hymeglusin for 24 h. We rationalized that simvastatin would induce protein changes that were due to both HMG-FAS and reductions in cholesterol metabolism, and that hymeglusin would induce changes that were due to only reductions in cholesterol metabolism; hymeglusin inhibits the same pathway but without inducing HMGylation of FAS (Fig. 3a). Lysates were collected and LC-MS/MS was performed to determine abundance levels of specific proteins in the various treatments relative to the control condition. Proteins with a p-value less than 0.05 were sorted into increased or decreased during statin treatment and increased or decreased during hymeglusin treatment. The proteins in these four groups were then used to identify pathway changes specific to each treatment. As simvastatin treatment will cause broad changes related to both inhibition of FAS and inhibition of the cholesterol biosynthetic pathway, hymeglusin treatment allowed us to sort out changes that occur in both conditions from inhibited cholesterol metabolism.

As a positive control, the "statin increase" and "hymeglusin increase" overlap had 11 commonly enriched pathways, all of which are related to the known increase in proteins of the cholesterol and isoprenoid pathways (Fig. 6f). When looking for changes unique to cells containing HMGylated FAS, the most consistently different pathways were related to ER and Golgi transport. Cells treated with hymeglusin had 19 unique pathways that were decreased, of these, 9 are related to the ER or Golgi. Conversely, statin-treated cells had no decreases in ER or Golgi pathways but had two related pathways that were increasing. Lastly, of the three pathways that were found to both increase in statin-treated cells, and decrease in hymeglusin-treated cells, one

is related to COPII vesicle formation, an important component of ER to Golgi transport (Fig. 6f and Supplementary Fig. 9a). Beyond ER and Golgi pathway enrichments, we also noticed several proteins either require binding of calcium or are otherwise regulated by it. This is notable because previously a subpool of FAS was shown to be interacting with SERCA and regulating its ability to reuptake calcium into the ER[13]. The placement of a modified pool of FAS next to the ER would give it spatial access to the composition of ER membrane as well as the proteins embedded within it and provide a mechanism for altering multiple processes. Together, these data show several pathways are altered by HMG-FAS, and suggest specific communication between lipid and cholesterol biosynthesis during statin treatment and broad cellular effects.

**HMGylation of FAS occurs during metformin-induced AMPK activation.** Finally, we considered whether this modification was a unique effect of statins or could be more broadly applicable. Beyond statin treatment, inhibition of HMGCR also occurs through phosphorylation by 5′ adenosine monophosphate-activated protein kinase (AMPK). AMPK is typically activated during times of energy stress and results in the phosphorylation and inhibition of several anabolic enzymes that are counterproductive during times of stress. AMPK is also activated by metformin, another widely prescribed drug.

Thus, we investigated whether activating AMPK would result in the accumulation of HMG-CoA through the inhibition of HMGCR, and subsequent modification of FAS. HepG2 cells were treated with metformin for 24 hours and revealed an increase in HMG-CoA levels (Fig. 7a). Similar to statin-treated cells, HMG-CoA was the only CoA to increase; however, several CoAs decreased, which is consistent with known AMPK effects[14] ($p < 0.01$, Fig. 7a).

Next, we performed western blotting and found that metformin treated cells had a singular HMGylation band occurs around 250 kDa, the same as during statin-treatment (Fig. 7b). Cells treated with another AMPK activator, AICAR, reveal similar results supporting the role of AMPK in the HMGylation of FAS (Supplementary Fig. 10).

We also looked to see if any cellular signaling changes occurred during metformin-treatment that overlapped with those found upon HMGylation of FAS. We included a set of metformin treated cells in our label-free quantitative proteomics to compare against statin- and hymeglusin-treated cells. Similarly, we filtered AMPK protein changes to identify enriched pathways with increased and decreased proteins, before comparing these pathways to statin/hymeglusin treatment. By looking for pathways that changed similarly between statin and metformin treated cells but not hymeglusin-treated cells, we were able to narrow the previous pathway list. Remarkably, of the 5 increased pathways that overlap between statin and metformin, 4 of these are related to membrane and vesicle transport (annotated as "neutrophils") (Fig. 7c and Supplementary Fig. 9b). These data provide further evidence that the HMGylation of FAS could be driving a change in ER and Golgi membrane transport and/or function.

## Discussion

The on-going discoveries around new classes of acyl-CoA modifications have revealed a novel means for metabolites to interact with their surroundings and modify cellular functions. Here we have shown that the inhibition of HMGCR leads to an increase in its substrate and the specific modification of FAS on active site residues. The modification's unusual attachment via ester and thioester bonds provides for rapid on and off loading of the HMG-modification without enzymatic intervention. The modification clearly impacts

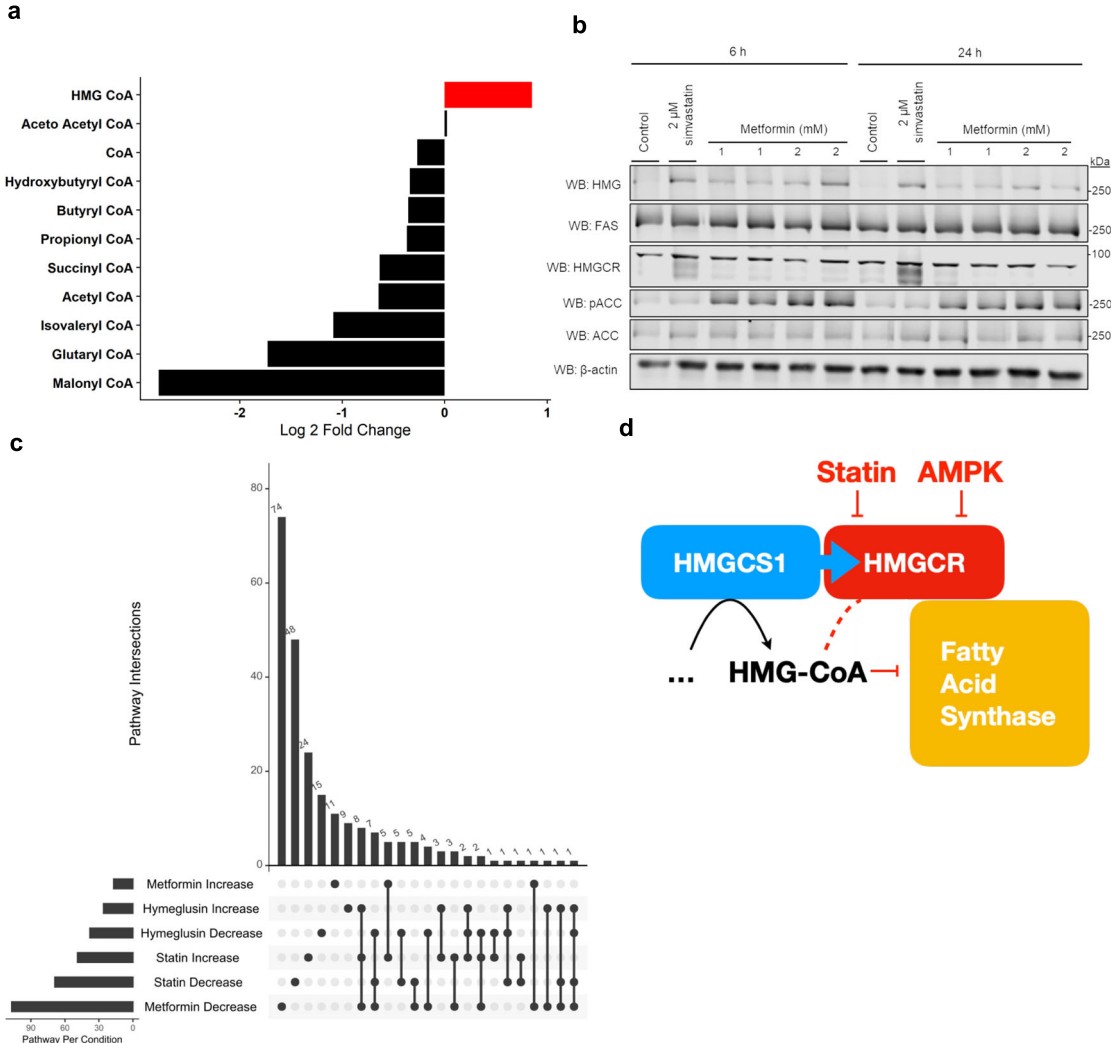

**Fig. 7 HMGylation of FAS occurs upon AMPK activation.** AMPK activation phosphorylates HMGCR to inhibit it, resulting in HMGylation of FAS and similar cellular signaling changes. **a** HepG2 cells treated with metformin, a known AMPK activator, show increased levels of HMG-CoA ($p = 0.0039$, $n = 3$ biological replicates, ANOVA followed by Tukey post-hoc test). **b** HepG2 cells treated with metformin show HMGylation of FAS in a similar manner to statin treatment. Phosphorylated ACC confirms AMPK activation during HMGylation of FAS; representative image shown from four independent experiments. **c** Upsetter plot showing which pathway changes overlap between treatment conditions. **d** Schematic showing the inhibition of HMGCR resulting in increased levels of HMG-CoA that modify the local pool of FAS. Source data are provided as a Source Data file.

the activity of FAS during in vitro experiments but surprisingly does not impact overall de novo lipogenesis when looking in an in vivo setting. With the discovery of the interaction between FAS and HMGCR, we propose that the inhibition is localized to FAS near the ER membrane and leads to signaling changes that may impact the ER-Golgi homeostasis (Fig. 7d). Furthermore, we have shown that this occurs during two of the most widely prescribed drug treatments.

In addition to the unusual specificity to modify FAS, the modification is also different from previously described acyl-modifications. Canonically, acyl modifications are linked to lysine residues via an amide bond; however, we found HMG-modifications on FAS that modify an active site serine as well as a phosphopantetheine prosthetic group. One key difference between these modifications and lysine modifications is their stability. While lysine modifications require an enzyme to break the stable bond, modifications via ester and thioester bonds are less stable and can be removed through stochastic chemical reactions. This is made clear with the displacement of the modification by malonyl-CoA (Fig. 4a, b), an event that would not

occur with an HMGylated lysine. The impact of this finding is the ability for a cellular system to rapidly regulate an enzyme and pathway using acyl modifications.

The exclusive increase in HMG-CoA and none of its precursors indicate that inhibition of metabolic enzymes can have specific impacts on metabolite pools. Interestingly, the identification of a single protein being modified reveals a new model in non-enzymatic protein acylation. Previous research has shown that genetically knocking out an enzyme that handles a reactive CoA leads to numerous acylation events on numerous proteins. The specific modification described here shows coordination and reveals a new means of communication between the de novo lipogenesis and cholesterol biosynthesis pathways. It is unlikely that this is the only pathway that would utilize accumulated acyl-CoA substrate to modify nearby proteins and further propagate a signal. By inhibiting one enzyme and accumulating an acyl-CoA, the cell can modify multiple enzymes and rapidly reverse these effects by altering only one input.

The discovery of the FAS and HMGCR interaction suggests there is an undescribed relationship between the two proteins and

pathways. The location of HMGCR on the ER membrane places it in an area that is both sensitive to lipid fluctuation and cellular stress allowing the ER to participate in cellular signaling and regulation. However, when identifying the proteins that are primarily responsible for driving these pathways enrichments, no clear unifying protein class or function was observed to prompt a concise follow up to test ER or Golgi functionality. Importantly, FAS activity has previously been shown to modulate membrane functionality. For example, knocking-out of FAS in macrophages prevents formation of microdomains that are required for protein recruitment, and subsequently prevents Rho GTPases from activating JNK[15]. While previous studies suggested that statins impact Golgi function, this effect is often attributed to decreased prenylation of Rho GTPases[16]. However, any decrease in prenylation should be accounted for by comparing statin and hymeglusin-treated cells, just as cholesterol pathway changes are identified. Furthermore, macrophages with FAS knocked out prevented proper Rho GTPase localization without impacting their prenylation status, indicating FAS plays a role in protein localization independent from prenylation modifications[15]. Furthermore, we found that several pathway changes overlap in statin and metformin treated cells that inversely change in hymeglusin-treated cells (Fig. 7c). These pathways imply a change in cellular signaling events centered around the ER and Golgi. Further work is required to confirm a connection and identify a mechanism between the HMGylation of FAS and any impacts of the function of the ER and Golgi.

Overall, we have identified a novel interaction between two lipid pathways through the HMGylation of FAS by the inhibition of HMGCR. This HMGylation appears to affect the ER and Golgi in unknown ways. As new methods and technologies are developed, it may become possible to better target the subpools of FAS to disentangle the effects of HMGylation from the effects of inhibition to the cholesterol pathway. Additional investigations into other pathways that use this mechanism could lead to new methods for identifying the effects of transient acyl-PTMs and the identification of novel cellular signaling pathways.

The discovery of acyl modifications occurring on serine residues and prosthetic groups could expand discoveries in the acyl-PTM field. These labile modifications can be missed by the current methods of detecting PTMs. Indeed, we identified several key steps that are important for identifying non-lysine acyl-PTMs, including the development of antibodies that will detect PTMs bound to residues other than lysine, as well as omitting heating and reducing agents from western blot preps. In addition to the above requirements, mass spectrometry requires the user to direct the program to look for modifications that are bound to residues other than lysine. These new discoveries will require redevelopment of commonly used methods as well as the reexamination of prior acyl-PTMs research to see if there are additional changes being conveyed through non-lysine modifications. An exciting future of non-lysine acyl-PTMs awaits discovery.

## Methods

**Cell lines and reagents**. HepG2 cells (ATCC) were cultured in surface treated flasks or dishes (Corning) in high glucose, no glutamine Dulbecco's modified Eagle medium (Thermo 11960) + 10% FBS (Thermo) at 37 °C with 5% CO$_2$. Cells were treated for the time indicated with simvastatin (Sigma) that was hydrolyzed as described previously[17]. Hymeglusin (Cayman Chemical) was dissolved in DMSO and added to media at 1:1000. Metformin (Sigma) was prepared fresh and dissolved directly in media. All conditions had DMSO added to the same concentration (0.1%). Cells were placed on ice before media was aspirated and cells washed twice with cold PBS before RIPA buffer (50 mM Tris pH 7.4, 1% NP-40, 0.5% sodium deoxycholate, 0.1% SDS, 150 mM NaCl, 1 mM EGTA) with protease inhibitors (Thermo) were added. Cells were scraped and lysate transferred to 1.7 mL tubes and centrifuged at 14,000 x g for 10 min at 4 °C. Supernatant was transferred to clean tube and either kept on ice or stored at −80 °C. Protein concentration was determined using bicinchoninic acid (BCA) method[18].

**Statin-treated mice**. All animal procedures were performed in accordance with the Association for the Assessment and Accreditation of Laboratory Animal Care (AAALAC) international guidelines and approved by the Duke University Institutional Animal Care & Use Committee under protocol number A065-20-03.

C57/Bl6NJ Mice were obtained from Jackson Labs and received ad libitum PicoLab rodent diet 20 (#5053, LabDiet, St. Louis, MO) and water. About 2–4 animals per cage were housed together with a 12 h light–dark cycle. All animal procedures were performed in accordance with the Association for the Assessment and Accreditation of Laboratory Animal Care (AAALAC) international guidelines and approved by the Duke University Institutional Animal Care & Use Committee. Both male and female mice from 12–24 weeks of age were used in experiments to account for potential differences due to sex and litter mates used in experiments to control for age differences.

At the time of treatment, control mice were given blended chow while treated mice received blended chow with 0.01% atorvastatin powdered and mixed in. Chow was replaced every 2–3 days and weights monitored to ensure there were no changes in weight between treatment groups. Mice were on diet for 12 days before administered 27 μL sterile isotonic deuterium/mg body weight via IP injection. At this time water was replaced with water containing 4% deuterium. Blood collections were taken via tail vein bleed, 30 min, 19 h, 26 h, and 48 h post deuterium injection to ensure mice had similar levels of deuterium enrichment. Two hours prior to endpoint, mice were gavaged with 100 mg atorvastatin/kg body weight. This was to ensure inhibition of HMGCR at the time of collection. Fourteen days following chow treatment, and 48 h post deuterium administration, mice were euthanized and both liver and skeletal tissue were collected and clamp frozen using a clamp chilled with liquid nitrogen (LN2). Samples were placed in LN2 before being ground using a motor and pestle chilled in LN2. Powdered samples were then stored at −80 °C for future analysis.

**Measurement of Acyl-CoA**. Acyl-CoA measurements were made using a previously described method[19]. HepG2 cells were plated in 10 cm plates (one million cells each) and cultured for 24 h. Cells were then treated with either 0.1% DMSO, 2 μM simvastatin, 2 μM hymeglusin, 2 μM simvastatin + 2 μM hymeglusin, or 2 mM Metformin. 24 h later cells were washed twice with cold PBS, before 1 mL of 0.3 M Perchloric acid was added. Plates were scraped and lysates transferred to 1.5 mL tube on ice. Stored at −80 °C. For mouse tissue preparation, ~65 mg of frozen liver was weighed out before adding 0.3 M perchloric acid at a ratio of 1 mg tissue/19 μL acid. A glass bead was placed in the tube and samples were disrupted using a Tissuelyser (Qiagen) set to 30 Hz for 2 min. Samples were transferred to a clean tube leaving the bead behind before being stored at −80 °C. The extracts were spiked with $^{13}$C2-Acetyl-CoA (Sigma, MO, USA), centrifuged, and filtered through the Millipore Ultrafree-MC 0.1 μm centrifugal filters before being injected onto the Chromolith FastGradient RP-18e HPLC column, 50 × 2 mm (EMD Millipore) and analyzed on a Waters Xevo TQ-S triple quadrupole mass spectrometer coupled to a Waters Acquity UPLC system (Waters, Milford, MA).

**Western blotting**. Thirty micrograms of of lysate were mixed with non-reducing sample buffer (0.2 M Tris pH 6.8, 50% Glycerol, 10% SDS, 0.01% Bromophenol blue) or reducing sample buffer (0.5 M DTT added). Samples were mixed but not heated unless otherwise stated, before being loaded onto a 7% Tris gel and separated by sodium dodecyl sulfate polyacrylamide gel electrophoresis (SDS-PAGE). The proteins were then transferred to a nitrocellulose membrane using a Bio-Rad wet transfer system. Membranes were then blocked with 1% fish gelatin (Sigma) in PBS for 1 h at room temp. Membranes were incubated overnight at 4 °C with primary antibody diluted in blocking buffer mixed with TBST 1:1. Additionally, for HMGylation blots, BSA at 30 mg/mL was added to the antibody working stock in order to limit off target binding. Membranes were then washed three times with TBST before being incubated with secondary antibody for 1 h at room temp. Membrane was washed three times with TBST before being imaged using an Odyssey CL-x imager (LI-COR). Antibodies and ratios used were HMG at 1:1000 (Millipore ABS2108), FAS at 1:5000 (abcam ab184619), HMGCR at 1:1000 (Millipore ABS229), Phosphorylated ACC at 1:1000 (CST 3661), total ACC at 1:1000 (CST 3662). Secondary antibodies were LI-COR 926-32213 donkey against rabbit or LI-COR 925-68072 donkey against mouse and used at 1:10000.

**Purification of FAS**. The pig mammary gland was a kind gift from Tim Haystead (Duke), originally obtained from Laurie Rund (Univ. Illinois at Urbana-Champaign). Tissue was collected from pigs maintained at a specific pathogen-free clean herd at the University of Illinois at Urbana-Champaign, which maintains a full Association for the Assessment and Accreditation of Laboratory Animal Care International (AAALAC) accreditation. Animal housing and husbandry practices were fully compliant with the Edition of the Guide for the Care and Use of Laboratory Animals (National Research Council, 2011) and with the 3rd Edition of the Guide for the Care and Use of Agricultural Animals in Research (Federation of Animal Science Societies, 2010). Briefly, we removed 2 g of lactating pig mammary gland and crushed tissue while still frozen. It was then transferred to glass tube and added 15 mL of Buffer (10 mM Sodium Phosphate Buffer pH 7.5, 100 mM NaF, 5 mM EDTA, 5% Glycerol + protease inhibitors (Pierce)). Next, it was homogenized using Potter-Elvehjem PTFE pestle spinning at 1 k rpm for 2.5 min,

incubating on ice 1 min, and homogenizing again for 2.5 min. We then added remaining 5 mL of buffer and transferred to ultra-speed centrifuge tubes. Centrifuged at 142,000 x $g$ for 45 min at 4 °C. Sample was delipidated by passing through 5 mL glass wool in a 30 mL syringe. Next we added ammonium sulfate solution dropwise to reach 20% and let stir at room temp for 20 min before placing on ice for 1 h. After centrifuging at 26,000 x $g$ for 20 min, we transferred the supernatant to flask and added ammonium sulfate to reach 35% and incubated for 20 min, incubated on ice 1 h. After centrifuging at 26,000 x $g$ for 20 min, we collected supernatant and washed pellet with buffer before resuspending each tube with 1 mL buffer. We concentrated ~5x using a 150 kDa MWCO filter centrifugation tube (Sigma). We added equal volume 100% glycerol to one aliquot mixing well before storing 50% glycerol stock at −20 °C. FAS concentration was determined by separating a sample using SDS-PAGE on a 7% Tris gel. Stained with InstantBlue (C.B.S. Scientific) for 1 h before imaging on Odessey CLx and quantifying the intensity of FAS band against BSA controls.

**In vitro chemical treatments of HMGylated FAS**. HMG-CoA (Sigma) and malonyl-CoA (Sigma) were dissolved in water before being aliquoted and stored at −80 °C until use. One microgram of purified FAS per reaction was incubated with various amount of HMG-CoA (10 μM was used unless otherwise indicated) for 30 min at 37 °C.

For time competition between HMG- and malonyl- CoA, 10 μM of HMG-CoA was mixed with various amounts of malonyl-CoA (0, 1, 5, 10, 25, 50, 100 μM). 1 μg of purified FAS per reaction was incubated for 10 min at 37 °C before being frozen on dry ice and stored at −80 °C. Western blotting was performed to detect HMGylation status.

Alternatively, 10 μM of HMG-CoA was incubated with purified FAS (1 μg/reaction) for 30 min at 37 °C. Malonyl-CoA was then added to the reaction to reach 10 μM and aliquots removed after 1, 5, 10, 15, 30, 45, 60 min and frozen on dry ice. Western blotting was performed to detect HMGylation status.

HepG2 cells were cultured in two T75 flasks (2 million cells per flask). After 24 h, one flask was treated with media containing 2 μM simvastatin. Twenty-four hours later cells were trypsinized and washed before resuspending cell pellet in RIPA buffer. Protein concentration was determined using BCA assay.

Four aliquots of 225 μg of statin-treated HepG2 lysate (30 μg x 7.5) were made and each mixed with a 4 M treatment solution (to yield 1 M final). Treatments were Tris pH 7.0, Hydroxylamine pH 7.0, CHES pH 9.6, and hydroxylamine pH 9.6. Aliquots were taken from each sample at 5, 10, 15, 20, 30, 45, 60 min post-treatment, added to sample buffer, and frozen on dry ice. Positive and negative controls (statin-treated lysate and control lysate) were taken at the start of the time course. Western blotting was used to reveal changes in HMGylation status.

Thirty micrograms of lysate from statin-treated cells was added to non-reducing sample buffer or sample buffer containing DTT (0.1 mM final) as indicated. Samples were then heated to 95 °C for 10 min or left at room temperature, as indicated. Heated samples were cooled to room temperature and spun down to collect all of the sample. Western blotting was performed as described above.

**Identification of HMGylation sites via mass spectrometry**. Two separate sample sets (control and HMG incubated) were prepared in parallel, one to be digested with Glu-C (Promega) for coverage of phosphopantetheine site and one digested with pepsin (Promega) for broad coverage of FAS active sites. Additionally, pepsin is active at low pH, which improves the stability of thioester bonds. Twenty-five micrograms of purified FAS had detergent removed using Pierce detergent removal columns (87777) and eluted in 50 mM Phosphate Buffer pH 7.0. Samples were then split in two and one was incubated with 10 μM HMG-CoA for 30 min at 37 °C while the other was incubated with water. From this point on, samples were kept at or below room temperature in an effort to preserve the modification. Samples were then incubated with 10 mM tris(2-carboxyethyl)phosphine (Sigma, TCEP) for 30 min at room temperature. TCEP was chosen for its selective reduction of thiols over thioesters as a method for preserving any $S$-HMG modifications[20]. Samples were then alkylated by incubating with 20 mM N-Ethylmaleimide for 25 min at room temperature.

Samples were then split again to create four reactions (-HMG/pepsin, + HMG/ pepsin, -HMG/GluC, + HMG/GluC). Pepsin samples were acidified to pH < 3 by adding 1% trifluoroacetic acid (TFA), 5% MeOH. Pepsin was added at a ratio of 1:100. GluC was added at a ratio of 1:20 with the addition of 0.5 mM Glu-Glu (Sigma). All samples were incubated overnight at room temperature, shaken at 500 RPM. GluC samples had pH reduced to 2 using 10% TFA. All samples were frozen at −80 °C.

The samples were thawed at room temperature and four microliters from each sample were injected for LC-MS/MS analysis on a nanoAcquity UPLC (Waters) coupled via electrospray ionization to a Q Exactive Plus Orbitrap high-resolution accurate mass tandem mass spectrometer (Thermo). The peptides were first trapped on a Symmetry C18 300 × 180 mm trapping column (5 μL/min at 99.9/ 0.1 v/v H$_2$O/MeCN). The analytical separation was then performed using a 1.7 μm Acquity HSS T3 C18 75 × 250 mm column (Waters) and a 60-min gradient of 5 to 40% MeCN (Fisher, LCMS grade) with 0.1% formic acid (Sigma) at a flow rate of 400 nL/min with a column temperature of 55 °C. Data collection on the Q Exactive Plus mass spectrometer was performed in a data-dependent MS/MS manner, using a 70,000 resolution precursor ion (MS1) scan followed by MS/MS (MS2) of the top

10 most abundant ions at 17,500 resolution. MS1 was accomplished using AGC target of 1e6 ions and max accumulation of 60 ms. MS2 used AGC target of 5e4 ions, 60 ms max accumulation, 2.0 $m/z$ isolation window, 27 V normalized collision energy, and 20 s dynamic exclusion. The total analysis cycle time for each sample injection was ~80 min.

After observation of the hydroxymethylglutaryl-phosphopantetheine (HMG-Ppant) post-translational modification on S2157 of the FAS protein, a parallel reaction monitoring (PRM) experiment was performed. Twenty-three peptides were chosen for monitoring at either + 2 or + 3 charge state or both along with the peptide containing the HMG-Ppant modification of interest for a total of 30 precursors of the custom method. The precursor $m/z$, charge state, and peptide sequence are shown in the source data file (Fig. 5 directory). The results of this experiment were compiled into Skyline 4.2.1.19004 for further analysis.

Two T75 flasks of HepG2 cells were cultured, one with standard media and one with 2 μM simvastatin added. Cells were incubated for 48 h before media was aspirated, cells washed with PBS, and trypsin added. Once cells were detached, additional media was added before collecting and pelleting cells. Media was aspirated, cells washed with PBS, and pelleted. PBS was aspirated and 250 μL of IP lysis buffer was added (50 mM Tris pH 7.4, 150 mM NaCl, 0.5 mM EDTA, 1% Triton X-100). Each pellet was sonicated for 10, one second bursts before being centrifuged at 14,000 x $g$ for 10 min at 4 °C. Following BCA assay, 600 μg of FAS was added to prepared FAS antibody-bead mixture and rotated for 1 h at 4 °C. Beads were magnetized and washed once with TBST and once with 50 mM PBS. PBS was added again and transferred to a clean tube. One more wash with PBS was performed before beads resuspended with PBS and stored on ice.

To each sample, a solution of 10% sodium dodecyl sulfate (Sigma, SDS) in 100 mM phosphate buffer (Sigma) was added to yield a 5% SDS concentration. The samples were incubated at 25 °C for 30 min at 600 rpm. The sample beads were magnetized to the side of the tube with a magnetic tube rack and allowed to incubate for 5 min at room temperature.

The supernatant was removed and transferred to a new, labeled 2.0 mL protein low-bind tube (Eppendorf), which was also allowed to incubate for 5 min at room temperature in the magnetic tube rack, to ensure no magnetic beads were transferred along with the supernatant. Five-hundred millimolar TCEP in 50 mM phosphate buffer was added to each sample to yield 10 mM in solution and incubated at 25 °C at 600 rpm for 30 min in a Thermomixer (Eppendorf). Five-hundred millimolar N-ethylmaleimide (Alfa Aesar, NEM) in 50 mM phosphate buffer was added to each sample to yield 20 mM in solution and incubated at room temperature in the dark for 30 min. Twelve percent phosphoric acid (Fisher, LC grade) was added to each sample to yield a 1.2% phosphoric acid concentration in solution. 90% methanol (Fisher, LC grade, MeOH) in 100 mM triethylammonium bicarbonate (Sigma, TEAB) (S-Trap binding buffer) was added to yield a 7x dilution of each sample.

The samples were divided by pipetting half the volume into a new, labeled 1.5 mL protein low-bind tube, creating four samples total. The sample volume was then pipetted onto the S-Trap tips (Protifi) into 200 μL aliquots and spun down for 10–20 s as needed in a benchtop centrifuge. Each sample was washed four times by adding 150 μL of S-Trap binding buffer to each S-Trap tip, and the eluent was discarded.

A vial containing sequencing grade Glu-C (Promega) was resuspended in 2 mM glu-glu (Sigma) and 50 mM phosphate buffer at a concentration of 30 ng/μL. Twenty microliters (600 ng) Glu-C was added to each S-Trap tip for the Glu-C statin and control samples. The samples were loosely capped and allowed to incubate overnight without shaking at 25 °C. A solution of 15 ng/μL pepsin was prepared in 0.1% TFA in 5% MeOH and 20 μL (300 ng) pepsin was added to each S-Trap tip for the pepsin statin and control samples. The samples were loosely capped and allowed to incubate for 4 h without shaking at 25 °C.

At the end of each respective sample incubation time for digestion, 40 μL of 50 mM phosphate buffer was added to the S-Trap tips and eluted along with sample peptides using the benchtop centrifuge at 5000 x $g$ for 10–20 s as needed. Forty microliters of 0.2% aqueous formic acid (Thermo, high purity ampules) was added to the S-Trap tips and peptides were eluted using the benchtop centrifuge as above. A final peptide elution was done by adding 35 μL of 50% acetonitrile in 0.2% formic acid to the S-Trap tip and spinning down using the benchtop centrifuge as above. The peptide eluents for each sample were dried on the Vacufuge (Eppendorf) at room temperature.

The samples were resuspended in 20 μL of 1/2/97 v/v/v TFA/acetonitrile/water and 4.0 μL was injected for analysis by LC-MS/MS on a nanoAcquity UPLC (Waters) coupled via electrospray ionization to an Orbitrap Fusion Lumos high-resolution accurate mass tandem mass spectrometer (Thermo). The peptides were first trapped on a Symmetry C18 180 μm x 20 mm trapping column (5.0 μL/min at 99.9/0.1 v/v H$_2$O/MeCN) for 5 min. The peptides were then eluted over a 60-min gradient of 5 to 40% acetonitrile at a flow rate of 400 nL/min with a column temperature of 55 °C. Data collection on the Orbitrap Fusion Lumos was performed in a data-dependent MS/MS manner, using a 120,000 resolution precursor ion scan with the orbitrap (MS1) followed by MS/MS (MS2) using the ion trap and the top 20 most abundant ions at 30,000 resolution. The MS1 scan was performed using an AGC target of 2e5 ions and max accumulation of 50 ms. The MS2 detection was performed using an AGC target of 5e3 ions and max accumulation of 100 ms including automatic peak detection capability, 2.0 $m/z$

isolation window and 27 V normalized collision energy. The total analysis cycle time for each sample injection was ~80 min.

The data was searched on Mascot v 2.5 using a SwissProt database with human taxonomy selected. The peptide tolerance was +/− 5 ppm and the product tolerance was +/− 0.8 Da. The variable modifications selected include phosphopantetheine (ppant)(S), ppant-NEM(S), ppant-HMG(S). oxidation (M), and deamidation (NQ). The data from the pepsin samples were searched with no enzyme cleavage rules selected. The data from the Glu-C samples were searched with Glu-C enzyme cleavage rules selected and 6 missed cleavages.

After observation of the hydroxymethylglutaryl-phosphopantetheine (HMG-Ppant) post-translational modification on S2156 of the FAS protein, a parallel reaction monitoring (PRM) experiment was performed. Five peptides were chosen for monitoring at either + 2 or + 3 charge state or both along with the peptide containing the HMG-Ppant modification of interest for a total of 11 precursors in the custom method. The precursor $m/z$, charge state, and peptide sequence are shown in the source data file (Fig. 5 directory). The results of this experiment were compiled into Skyline 4.2 for further analysis.

**Modeling of HMGylation sites on FAS.** The crystal structure coordinates for mammalian fatty acid synthase were downloaded from the Protein Data Bank (www.rcsb.org, PDB ID: 2VZ9)[21]. As the C-terminus of the structure did not contain the phosphopantethine-modified Ser2157, we utilized YASARA Structure v19.12 (YASARA Biosciences GmbH, Vienna, Austria) to add the missing C-terminal residues through Glu2170. The phosphopantethine and HMG modifications were built onto the structures using Biovia Discovery Studio 2020 (Biovia, Inc, San Diego, CA). The final structures were then subjected to explicit solvent-based energy minimization in YASARA using the AMBER14 forcefield[22]. Images were rendered using Lightwave 2019.3 (Lightwave3D Group, Burbank, CA).

**Fatty acid synthase activity assay.** FAS activity assay was adapted from prior work[23]. In all, 30 nM (final concentration) FAS was incubated with water or the indicated amount of HMG-CoA for 30 min at 37 °C prior to adding to reaction mixture. Reaction mixture final concentrations were 50 mM Phosphate pH 7.0, 40 μM Acetyl-CoA (Sigma), and 60 μM NADPH (Roche) and final volume of the reaction was 300 μL. Reactions also had HMG-CoA added to match the pre-incubation levels. Malonyl-CoA (Sigma, final 40 μM) was added to start the reaction. Samples without malonyl-CoA were used to calculate the background signal. Activity was determined by using UV–Vis spectroscopy to measure the oxidation of NADPH. Measurements were taken at 340 nm every 20 s for 10 min. 1 s of shaking was performed prior to each read, and the reaction performed at room temperature. The slope of absorbance vs time was calculated for each sample and the background sample was subtracted from all sample before HMG-CoA containing samples were calculated as a percent of malonyl-CoA control samples.

**De novo *Lipogenesis* in cells.** HepG2 cells were plated in 6 cm dishes (one million per dish). Following 24 h incubation, media was changed to media + 10% deuterated water ± 2 μM simvastatin. Three replicates per treatment were collected at each time point (2, 4, 6, 18, 24 h). The cells were washed twice with chilled PBS. Metabolic activity was quenched by the addition of 1 mL of cold methanol (−20 °C) and cells were subsequently placed on ice. Cells were scraped and the cell suspension was transferred into glass tubes with Teflon-sealed caps. One milliliter of cold H₂O was added to the culture dish and transferred to the same tube. Next, 1 mL of chloroform was added, and the tubes were vortexed vigorously for 60 s. The tubes were stored overnight at 4 °C. The following day, the tubes were centrifuged at 5000 x g and 4 °C for 10 min. The resulting phase separation resulted in an aqueous upper phase containing polar metabolites and an organic lower phase containing nonpolar metabolites. The lower phase (chloroform) was transferred to a new glass tube with a Teflon-sealed cap. Samples were stored at −80 °C until further use.

For GC-MS analysis of fatty acid methyl esters (FAMEs), samples were prepared as previously detailed[24]. In short, the chloroform phase was evaporated to dryness under nitrogen at 37 °C. Then samples were treated for 2 h at 100 °C by acidic methylation (1 mL methanol and 50 μL sulfuric acid). FAMEs were isolated through hexane/water (2:1 v/v) extraction and the hexane phase was evaporated under nitrogen at 37 °C. Lastly, FAMEs were resuspended in 70 μL of hexane and transferred to a GC vial for MS-analysis.

GC-MS analysis was performed on an Agilent 7890B GC system equipped with a HP-5MS capillary column connected to an Agilent 5977 A Mass Spectrometer following the method of Long & Antoniewicz. Mass isotopomer distributions were obtained by integration of ion chromatograms[25] and corrected for natural isotope abundances. Metabolites monitored were myristic acid ($m/z$ 242–256); palmitic acid ($m/z$ 270–286); and stearic acid ($m/z$ 298–316). Isotopomer spectral analysis (ISA) model parameters[26] were estimated by minimizing the variance-weighted sum of squared residuals (SSR) between the experimentally measured and model predicted mass isotopomers of fatty acids using non-linear least-squares regression[27]. The fitted results were subjected to a $\chi^2$ statistical test to assess the goodness-of-fit, and accurate 95% confidence intervals were computed for all estimated parameters by evaluating the SSR sensitivity to flux variations[28,29].

**²H-isotopomer spectral analysis.** A three-parameter ISA model was used to model the MID of palmitic acid obtained from deuterated water tracing experiments. In addition to the D and g-parameters, the number of water exchangeable hydrogen atoms (n) must be determined. The D parameter for the model was fixed at 0.1. As palmitate MIDs were obtained across five time points (2, 4, 6, 18, 24 h), we modeled the data in parallel as previously described[30–32], where each MID (for a given time point) had a unique g-value, but the same "n" was regressed for all data sets. For in vitro studies, we found $n = 16$ as the best-fit for palmitic acid, which is in the range of previous studies[33–35]. The fractional synthesis rate was then determined by regressing a best-fit through the g(t) vs. time data.

**De novo *Lipogenesis* in C57Bl/6NJ mice.** Sample and calibration curve preparations were performed as follows. The plasma ²H₂O labeling was assayed according to the published protocol[36]. Briefly, for standard curve ²H₂O preparation, we prepared 0, 0.1, 0.5, 1.0, 1.5, 2.2, 2.5, 3.0, 3.5, and 4.0% ²H₂O (Sigma-Aldrich, St. Louis, MO) in distilled water for calibration stock solutions. Ten microliters of plasma sample or standard solution was mixed with 2 μL 10 M NaOH and 4 μL acetone/acetonitrile (1:20, volume ratio, acetone: acetonitrile) solutions. Samples were vortexed and briefly centrifuged (~5 s, 14,000 x g) to ensure that all of the drops were at the bottom of the tube. Cap the tube and let the sample sit for overnight (at least for 10 h). A 500 μL chloroform was added to the above sample or standard solutions together with 500 mg Na₂SO₄ powder. Samples or standard solutions were vortexed and centrifuged at 800 × g for 2 min. Pipetted 100 μL of chloroform layer into GC-MS vial with glass insert for GC-MS analysis.

*Gas chromatography mass spectrometry, GC-MS (EI mode).* The extracted acetone was analyzed using an Agilent 5973N-MSD equipped with an Agilent 6890 GC system, and a DB-17MS capillary column (30 m x 0.25 mm × 0.25 μm). The mass spectrometer was operated in the electron impact mode (EI; 70 eV). The temperature program was as follows: 60 °C initial, increase by 20 °C/min to 100 °C, increase by 50 °C/min to 220 °C, and hold for 1 min. The sample was injected at a split ratio of 40:1 with a helium flow of 1 mL/min. Acetone eluted at 1.5 min. Selective ion monitoring of mass-to-charge ratios of 58 and 59 was using a dwell time of 10 ms/ion for M0 and M1 acetone.

The total palmitic acid labeling is assayed based on the published work[36]. Briefly 20 mg liver tissue was homogenized in 1 mL KOH/EtOH (EtOH 75%) and incubated at 85 °C for 3 h. A 200 μL of internal standard heptadecanoic acid (1 mg/mL) was added into samples after cool down. A 100 μL of sample was acidified by 100 μL of 6 M HCl. Palmitic acid was extracted by 600 μL chloroform. Chloroform layer was completely dried by nitrogen gas and was reacted with 50 μL N-methyl-N-trimethylsilylfluoroacetamide (TMS) at 70 °C for 30 min. The TMS-derivatized samples were ready for GC-MS analysis.

*Gas chromatography mass spectrometry, GC-MS (EI mode).* Palmitate–TMS derivative was analyzed using an Agilent 5973N-MSD equipped with an Agilent 6890 GC system, and a DB-17MS capillary column (30 m x 0.25 mm × 0.25 μm). The mass spectrometer was operated in the electron impact mode (EI; 70 eV). The temperature program was as follows: 100 °C initial, increase by 15 °C/min to 295 °C and hold for 8 min. The sample was injected at a split ratio of 10:1 with a helium flow of 1 mL/min. Palmitate–TMS derivative eluted at 9.7 min. Mass scan from 100 to 600 was chosen in the method. The $m/z$ at 313, 314, and 327 were extracted for M0/M1 palmitate and heptadecanoic acid quantification.

All the stable isotope labeling was corrected from the natural stable isotope distribution[37]. The newly synthesized total palmitic acid was calculated as following:[36] %newly synthesized palmitic acid labeling = total palmitic acid labeling /(plasma ²H₂O labeling × 22) × 100.

**Proximity ligation assay.** HepG2 cells were plated on 12 mM coverslips (Warner Instruments 64-0712) in 24-well plates and grown in standard media for 48 h. Cells were then washed with PBS followed by PHEM (60 mM PIPES pH 7, 25 mM HEPES, 10 mM EGTA, 2 mM MgCl₂) before being fixed with 3% formaldehyde in PHEM for 13 min at 37 °C. Cells were washed with PHEM before being permeabilized using 0.5% NP-40 in PHEM. Cells were washed in PBS before being stored in PBS at 4 °C.

PLA probes were made using Sigma Duolink Probemaker kit. HMGCR antibody from abcam (ab214018) was ligated to PLUS oligo and FAS antibody from abcam (ab18461) was ligated to MINUS oligo following Probemaker instructions. Cells underwent PLA reaction using Sigma Duolink In Situ Detection Reagents Green kit. In brief, cells were blocked using Duolink blocking reagent before being incubated with probe mixture. Mixtures contained FAS antibodies at a ratio of 1:50 and HMGCR antibodies at 3 μg/mL. Negative controls included samples with no probes, FAS probe only, and HMGCR probe only. Cells were washed and then incubated in ligation mixture before being washed again and incubated in elongation mixture. Cells were washed and incubated in 1 μg/mL Hoechst for 5 min at room temperature. Cells were washed once more before being mounted on slides using ProLong Gold (Thermo) and curing for 24 h.

Cells were viewed using a Leica DMI6000CS confocal microscope equipped with a HCX PL APO lambda blue 40x/1.25-0.75 oil objective and images captured using Leica LAS AF 2.7.3.9723 software. 405 Diode laser and Argon/2 laser were

used in wavelength ranges of 415–490 nm and 498–592 nm, respectively, with a pinhole of 67.9 μm. Sequential scanning at 400 Hz was used to image the cells with an image resolution of 1024 × 1024 pixels.

**Co-immunoprecipitation of HMGCR**. HepG2 cells grown in 10 cm dishes were washed with PBS and scraped into 0.5 mL ice cold lysis buffer (50 mM Tris HCl, pH 7.5, 250 mM NaCl, 1% Triton X-100, 2 mM EGTA, 2 mM EDTA, and HALT protease inhibitors). The lysate was incubated 30 min on ice, sonicated 3 min in an ice cold sonicator water bath at 10 Watts (RMS) output power, and centrifuged for 30 min at 18,000 x g. The resulting supernatant was added to HMGCR-conjugated agarose (Santa Cruz Biotechnology, sc-271595) or unconjugated agarose, for HMGCR and control IP s, respectively, and incubated overnight at 4 °C while rotating end-over-end. Beads were then washed 3x with lysis buffer and protein eluted by incubation of the beads with 80 μL elution buffer (non-reducing sample buffer) at room temperature for 10 min. Eluted protein was run on 4–15% poly-acrylamide gels and western blotting performed using methods listed above. Each blot was probed with anti-FAS mouse monoclonal (Sigma-Aldrich, WH0002194M1) at 1:1000 and anti-HMGylation rabbit polyclonal (made in-house) at 1:100.

**Label-free quantitative proteomics**. HepG2 cells were plated on 10 cm dishes with three replicates per treatment condition (0.1% DMSO, 2 μM simvastatin, 2 μM hymeglusin, 2 mM metformin) and were incubated for 24 h. Cells were trypsinized, washed with cold PBS, and stored at −80 °C as cell pellets.

Cell pellets were thawed on ice and 75 μL of freshly made 8 M Urea, 50 mM Tris pH 7.4, 40 mM NaCl buffer was added. Cells were incubated on ice for 5 min before sonication for 5 s each. Samples were centrifuged at 14,000 x g for 10 min at 4 °C. A BCA assay was performed to determine protein concentration. Fifty micrograms of protein from each sample was prepared in parallel and DTT was added to 5 mM and incubated for 30 min at 32 °C while shaken at 300 RPM. Iodoacetamide was added to 15 mM and incubated in the dark for 25 min at room temperature. Additional DTT was added to bring the final concentration to 15 mM. In all, 0.2 μg of LysC were added to each reaction and incubated at 32 °C for 3.5 h. 50 mM Tris pH 7.4 + 5 mM CaCl was added to reaction to dilute urea concentration to 1.5 M. One microgram of Trypsin was added to reaction and incubated 16 h at 32 °C. pH was reduced to below 2 with addition of TFA to 0.5% v/v concentration. Samples were vortexed and centrifuged at 10,000 x g for 10 min at room temperature. Solid Phase Extraction was performed using tC18 SEP-PAK columns (Waters). Samples were resuspended in 0.1% formic acid and peptide concentration was assed (Pierce Quantitative Colorimetric Peptide Assay).

Samples were subjected to nLC-MS/MS using an EASY-nLC UHPLC system (Thermo) system coupled to a Q Exactive Plus Hybrid Quadrupole-Orbitrap mass spectrometer (ThermoFisher Scientific) via a nanoelectrospray ionization source. QC pools were prepared by pooling an equal volume of all samples. Samples were run in a randomized order, with two 30 min blanks in between each sample, and a study pool QC every five experimental samples. Sample injections of 1 μg (~2 μL for each, sample-specific volumes based on peptide concentration) were first trapped on an Acclaim PepMap 100 C18 trapping column (3 μm particle size, 75 × 20 mm) with solvent A (0.1% FA) at a variable flow rate dictated by max pressure of 500 Bar, after which the analytical separation was performed over a 105 min gradient (flow rate of 300 nL/min) of 5 to 40% solvent B (90% ACN, 0.1% FA) using an Acclaim PepMap RSLC C18 analytical column (2 μm particle size, 75 × 500 mm column (Thermo Fischer Scientific) with a column temperature of 55 °C. MS1 (precursor ions) was performed with default settings of 70,000 resolution, an AGC target of $3 \times 10^6$ ions, and a maximum injection time (IT) of 100 ms. MS2 spectra (product ions) were collected by data-dependent acquisition (DDA) of the top 20 (loop count) most abundant precursor ions with a charge greater than 1 per MS1 scan with dynamic exclusion enabled for a window of 30 s. Precursor ions were filtered with a 1.2 $m/z$ isolation window and fragmented with a normalized collision energy (NCE) of 27. MS2 scans were performed at 35,000 resolution with an AGC target of $1 \times 10^5$ ions and a maximum IT of 100 ms.

Data analysis was performed using Proteome Discoverer 2.2. Sequest HT was used to search spectra against a target-decoy UniProt human complete proteome database of reviewed (Swiss-Prot) and unreviewed (TrEMBL) proteins, which consisted of 74,349 sequences on the date of download (4/6/18). The following default parameters: oxidation (15.995 Da on M) as a variable modification, carbamidomethyl (57.021 Da on C) as a fixed modification, and two missed cleavages (full trypsin specificity). Mass tolerances were 10 ppm and 0.02 Da for precursor and fragment ions, respectively. Percolator was used to filter PSMs to a 1% false discovery rate (FDR). PSMs were grouped to peptides maintaining 1% FDR at the peptide level and peptides were grouped to proteins using the rules of strict parsimony. Proteins were filtered to 1% FDR using the Protein FDR Validator. Peptide quantification was done using the MS1 precursor intensity, normalized to total peptide amount detected in each sample. Imputation was performed via low abundance resampling. Protein abundances were calculated as the median of ratios of all connected peptides, with an average CV of 22.3% for all proteins quantified across replicate injections of the Study QC Pool. Statistical significance was conducted with a t-test (Background Based).

Using R (4.0), we sorted proteins into lists based on treatment and fold change direction. Only proteins where $p < 0.05$ were included. Using the R package EnrichR, we then compared these protein lists against the 2018 Gene Ontology

database and identified what pathways were enriched in each list[38–41]. Next, we compared the overlaps and differences in changed pathways between each group by generating an "upset" plot using the R package UpSetR[42]. Each intersection of pathways was exported to an excel table for manual review of related pathways.

**Reporting summary**. Further information on research design is available in the Nature Research Reporting Summary linked to this article.

## Data availability

Source data are provided with this paper. Additionally, raw LC-MS/MS data used in Figs. 4, 5 and 6 have been deposited to the ProteomeXchange Consortium via the PRIDE partner repository with the dataset identifier PXD030952 [http://proteomecentral.proteomexchange.org/cgi/GetDataset? ID = PXD030952]. The FAS structure used for molecular visualization are available at the Protein Data Bank (www.rcsb.org) under the accession code 2VZ9. Metabolomics data have been deposited in Zenodo, along with all the other raw data, under https://doi.org/10.5281/zenodo.5819041.

## Code availability

R (4.0) was used to generate some of the statistical analyses and figures in this paper. All code can be found at https://github.com/hirscheylab/Statin-therapy-inhibits-fatty-acid-synthase-via-dynamic-protein-modifications. Additionally, the source code has been deposited in the Zenodo database at https://doi.org/10.5281/zenodo.6275667.

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

## Acknowledgements

We would like to thank the members of the Hirschey laboratory for their discussion and comments. In particular, we would like to thank Allie Mills, Ph.D and Brett Peterson, Ph.D for sharing their guidance. We would like to thank Timothy Haystead, Ph.D for providing the pig mammary gland and guidance in FAS purification, Duke University School of Medicine for the use of the Proteomics and Metabolomics shared resource as well as the Light Microscopy Core Facility. We would like to thank Dave Piston, Ph.D and Dan Foust for their efforts to identify changes in calcium flux following statin treatment. Grants sponsoring this work include The Glenn Foundation (M.D.H.), the National Institutes of Health/NIA grant R01AG045351 (M.D.H.), the National Institutes of Health/NIDDK grant R01DK115568 (M.D.H.), an NIH/NIGMS training grant to Duke University Pharmacological Sciences Training Program 5T32GM007105-40 (A.G.T.) and an NIH pre-doctoral fellowship 5F31HL139140 (A.G.T.), the National Institutes of Health/NIDDK training grant 2T32DK007012 (G.R.W.), and the North Carolina Diabetes Research Center (NCDRC) grant P30 DK124723 (S.B.C., G.F.Z., O.R.I., R.D.S.).

## Author contributions

Conceptualization, A.G.T, G.R.W., and M.D.H.; Investigation, A.G.T., G.R.W., K.A.A., J.W.T., S.B.C., R.D.S., P.A.G., O.R.I., D.S.B., G-F. Z., R.A.K, J.L.M.; Writing—original draft, A.G.T., M.D.H.; Writing—review & editing, All authors; Supervision, M.D.H.; Project administration, M.D.H.; Funding acquisition, M.D.H.

## Competing interests

The authors declare no competing interests.
