## [Peer Review File · Nature Communications]

REVIEWERS' COMMENTS

Reviewer #1 (Remarks to the Author):

I think the authors have responded well to the issues raised and I found their rebuttal to be persuasive. The gaps that could not be addressed for technical reasons were explained in a plausible manner and the role of a local pool of FAS etc seems possible. In some ways it's a pity that the data now in Fig 2 using HMG-S-NAC was not included in the original submission. The only minor point is that if this is the first report of the use of HMG-S-NAC in the literature they should really supply the full characterisation.

Reviewer #4 (Remarks to the Author):

The authors have thoughtfully and thoroughly addressed the concerns raised by the reviewers. This paper reports intriguing findings and should be published. I recommend acceptance.

Response to reviewer comments

(original, unmodified comments in blue, responses in black).

Reviewer #1 (Remarks to the Author):

I think the authors have responded well to the issues raised and I found their rebuttal to be persuasive. The gaps that could not be addressed for technical reasons were explained in a plausible manner and the role of a local pool of FAS etc seems possible. In some ways it's a pity that the data now in Fig 2 using HMG-S-NAC was not included in the original submission. The only minor point is that if this is the first report of the use of HMG-S-NAC in the literature they should really supply the full characterisation.

Thank you for the supportive comments. We now include a full characterization of HMG-S-NAC, including experimental details of its complete synthesis (methods), as well as characterization and validation of the compound by three independent NMR approaches (supplementary information)

Reviewer #4 (Remarks to the Author):

The authors have thoughtfully and thoroughly addressed the concerns raised by the reviewers. This paper reports intriguing findings and should be published. I recommend acceptance.

Thank you for the supportive comments.